



# An evaluation on the alignment of drought policy and planning guidelines with the contemporary disaster risk reduction agenda

Ilyas Masih

IHE Delft Institute for Water Education, Westvest 7, P.O. Box 3015, 2601 DA, Delft, the Netherlands

*Correspondence to*: Ilyas Masih (i.masih@un-ihe.org)

**Abstract.** Drought is a major global challenge causing significant socio-economic and environmental impacts. A paradigm shift from crisis to risk management is advocated to reduce the impacts of droughts, and to build the resilience of societies, and water and environmental systems against drought. A number of drought policy and planning guidelines are developed and used to support the transition from crisis to risk management and enhancing resilience. However, research is lacking on critical

reflection, evaluation and update of the available drought guidelines. For example, there is no study on assessing the correspondence of the available guidelines to the contemporary disaster risk reduction agenda. Therefore, this study evaluates twelve drought policy and planning guidelines for their alignment with the four priority areas of the SENDAI framework for disaster risk reduction 2015-2030. A qualitative evaluation matrix was developed and used in the assessment. The examined priorities and associated thematic elements were scored in the range 0-100, and classified under Very Low (0-10), Low (11-

30), Medium-Low (31-50), Medium-High (51-70), High (71-90), and Very High (91-100) categories. Most guidelines achieved (medium) high to very high scores on data and information, risk assessment, and communication and dissemination elements associated with priority 1 (understanding disaster risk). Whereas, mostly very low to low coverage was found for science-policy-practice dialogue, local knowledge and practices, and research and development. Strengthening disaster risk governance to manage disaster risk (priority 2) earned high scores on most elements, notably for strategies and plans,

coordination mechanisms, community representation, and policy and governance. In contrast, most elements under priority 3 (investing in disaster risk reduction) were classified under low to medium categories, which include financial allocation, risk transfer, and mainstreaming drought risk reduction into land use and rural development planning, business resilience and protection of livelihoods, and health and safety. Most elements under priority 4 (enhancing disaster preparedness) scored under medium low to medium high ranges, as sufficient information was lacking on multi-hazard early warning systems, post-disaster

recovery, rehabilitation, and reconstruction, and resilience of critical infrastructure. Furthermore, the study outlined several strengths, weaknesses, opportunities and threats pertaining to the examined guidelines. In general, the study reveals an urgent need to better align drought policy and planning guidelines with the contemporary disaster risk reduction agenda outlined in the SENDAI Framework. The findings of this study can be instructive in designing the next generation of drought guidelines in support of an accelerated transition towards drought risk management, and building resilient societies and ecosystems under

a changing climate and increasing anthropogenic pressures.



**Key words:** Drought risk management, Drought policy and plans, Drought guidelines, SENDAI Framework, SWOT analysis

## 1 Introduction

Drought is a major global challenge. Many countries face drought every year, and has to bear losses to a varying degree depending upon multiple factors such as drought severity and duration, geographical extent, vulnerability and resilience. There were 488 drought events recorded in the international disaster database (EM-DAT, 2024) during last thirty years (1994-2023) (Supplementary material 1). The estimates suggest that these droughts affected about 2 billion people across the globe, and caused a total economic damage of about 220 billion USD. A conservative estimate on the number of countries reporting drought in a year ranged from 6 (1995) to 29 (2015). The studies show that the drought events demonstrate local, regional, continental and global coverage (Masih et al., 2014; Blauhut et al., 2022; Mondal et al., 2023). Drought impacts (directly or indirectly) various sectors such as agriculture, water management, energy, river transport, environment, and public health (Wilhite et al., 2007; UNDRR 2021; Rossi et al., 2023). These impacts can be short, medium or long term, and may prevail over local, regional and global scales. For example, 2018-2020 drought event covered highest area in Europe (36 %) compared to previous droughts (Rakovec et al., 2022). The duration of this event was estimated at 12.2 months, and the event was estimated longest since last 250 years. The Central and Western European countries were most impacted by this drought. Moreover, the study stressed the need of adopting European policies, plans and strategies to cope with increasingly intense, long-duration and widespread droughts. This is also a global need because drought occurrences and impacts are most likely to increase in the future for many countries because of climate change (Spinoni et al., 2019; Naumann et al., 2021; Rakovec et al., 2022; IPCC, 2021 & 2022) and human activities (Van Loon et al., 2022).

A paradigm shift, in drought policy and practice, from crisis to risk management is advocated to reduce the impacts of droughts, and to build the resilience of societies, water and environmental systems against drought (Wilhite, 1991; Wilhite et al., 2000; World Bank, 2012; Sivakumar et al., 2014; UNISDR, 2005 & 2015; UNDRR 2021). Wilhite (1991) and Wilhite et al. (2000) proposed a novel Ten-Step Process to guide drought policy and planning process in support of transition towards risk management. The proposed approach was underpinned by the understanding and experience of science, policy and practice from the USA. Similar to the Ten-Step Process, MEDROPLAN drought guidelines were developed to support pro-active and risk-based approaches to address droughts in the Mediterranean region, which is highly vulnerability to droughts (Iglasias et al., 2007). Furthermore, the EU drought guidelines recommend an integrated and risk management approach, with strong focus on making drought plans at river basin level or integrating them as part of the river basin plans (European Commission, 2007). These guidelines also focus on drought management in relation to agriculture, climate change, transboundary cooperation, groundwater, sustainable development and environmental impact assessment. Furthermore, UNISDR (2007 & 2009) prepared a comprehensive drought risk reduction framework, which is aligned with the five priorities outlined in the Hyogo Framework for Action 2005-2015 (UNISDR, 2005). The UNISDR Framework is detailed around five key elements: 1) policy and governance; 2) drought risk identification and early warning; 3) awareness and education; 4) reducing underlying factors of





drought risk; and 5) mitigation and preparedness. Few years later, a high-level meeting on national drought policy (HMNDP)
was held in 2013 (Sivakumar et al., 2014). The final declaration of this landmark meeting note that drought poses a serious
challenge for the sustainable development of all countries, in particular developing countries. Many countries do not have
sufficient policies for appropriate drought management and pro-active drought preparedness, and drought responses are often
reactive (crisis management). The recent research corroborates this declaration, highlighting the variable degree of
preparedness and transition towards risk management within and across countries (Fu et al., 2013; Jedd et al., 2021; Blauhut
et al., 2022; Jedd and Smith 2023; Biella et al., 2024). Moreover, HMNDP recognized the urgent need of developing risk
management strategies and preparedness plans (Sivakumar et al., 2014), and the countries were encouraged to develop and
implement national drought management policies and plans. An invitation was extended to update the relevant science and
policy documents by aligning them to the recommendations made by HMNDP, which suggest to focus on developing pro-
active drought management measures, enhancing collaboration and quality of observation networks and delivery systems,
improve public awareness, consider suitable economic and financial strategies, establish emergency relief plans, and linking
drought management plans to local/national development policies. Following on the HMNDP recommendations, WMO and
GWP (2014) proposed national drought management policy guidelines, which are based on the Ten-Step Process (Wilhite
1991; Wilhite et al., 2000).

Building on the Hyogo framework, the SENDAI framework for disaster risk reduction 2015-2030 acknowledges the challenges
posed by multiple disasters, despite progress made during the past decades (UNISDR, 2015). The SENDAI framework presents
four priorities for action: Priority 1) Understanding disaster risk; Priority 2) strengthening disaster risk governance to manage
disaster risk; Priority 3) Investing in disaster risk reduction for resilience; and Priority 4) Enhancing disaster preparedness for
effective response, and to <<Build Back Better>> in recovery, rehabilitation and reconstruction. There are a few drought
guidelines developed after the SENDAI framework (UNCCD, 2018 & 2019; World Bank, 2019; Marj and Abadi, 2020; Filho,
F. A., et al., 2023). However, there is a lack of information on how these guidelines consider the goals and priorities of the
SENDAI framework, and related global disaster risk reduction agendas. Nevertheless, most recent guidelines (UNCCD, 2019;
World Bank, 2019) highlight the importance of three pillars of drought risk reduction (Tsegai et al., 2015; Verbist et al., 2016),
which include drought monitoring and early warning (pillar 1), vulnerability and impact assessment (pillar 2) and drought
mitigation and preparedness measures (pillar 3). These three pillars are a sub-set of the elements outlined as the priorities of
the SENDAI framework, and also reflect the components included in the declaration of HMNDP.

In general, there are a number of drought policy and planning guidelines developed before and after the SENDAI framework.
However, there is a lack of understanding on how the available guidelines align with the contemporary disaster risk reduction
agenda. Furthermore, adjusting drought policy and plans to the contemporary drought thinking and changing needs is essential
to accelerate progress on drought risk reduction and building resilience of societies against drought under changing climate
and increasing anthropogenic pressures. While several, global, regional and local guidelines are developed over the past fifty
years, the research is lacking on critical reflection, evaluation and update of these guidelines. To date, there is no study, to
author's knowledge, on assessing the correspondence of the available drought guidelines to the contemporary disaster risk





reduction agenda. Therefore, this study evaluates the drought policy and planning guidelines for their alignment with the four priority areas delineated in the SENDAI Framework for disaster risk reduction 2015-2030. Furthermore, the study explores

strengths, weaknesses, opportunities and threats, and provides insights for better aligning future generation of drought guidelines with the contemporary disaster risk reduction agenda.

## 2 Materials and Methods

The drought policy and planning guidelines were searched through multiple internet sources such as Scopus, Google and Google Scholar. The document search also benefited from the Author's knowledge gained through education and capacity

development activities related to drought management including teachings on the drought policy and planning guidelines. Various key words were used in finding the guidelines, which were mainly centred around drought policy, drought planning, drought guidelines, drought policy and planning framework, drought risk management. The search resulted in the selection of twelve guidelines published as journals articles and reports in English language. A brief description of these guidelines and main references for further details are provided in Table 1. Few more insightful documents and web sources were found (EDO,

2024; IDMP, 2024; NDMC Planning, 2024; Steinemann and Cavalcanti, 2006; UNISDR, UNDP and IUCN, 2009; Rossi et al., 2007; Rossi and Castiglione 2011; WMO and GWP, 2016; Cook et al., 2017; Vogt et al., 2018; CISA, 2021; Vogel and Kroll, 2021), but were not selected for evaluation because of their limited scope compared to this study's objectives, lack of details needed to conduct the evaluation or very high degree of similarity with the selected guidelines. It is acknowledged that the list of evaluated guidelines is not exhaustive, but sufficient for the purpose of this study.

**Table 1: A summary of the drought policy and planning guidelines evaluated in this study.**

| A brief description of the examined drought policy and planning guidelines, and suggested sources for further information |
|---|
| **The Ten-Step drought planning process (Wilhite, 1991; Wilhite et al., 2000; WMO and GWP, 2014; UNDRR, 2021)** |
| The Ten-Step Process is a ground-breaking work on drought policy and planning. This novel and most widely used process was developed based on the experience from the USA. The process is composed of ten-steps. Step 1: Appoint a national drought management policy commission to supervise and coordinate the policy and plan development and implementation at all levels of government. Step 2: State or define the goals and objectives of a risk-based national drought management policy. Step 3: Seek stakeholder participation; define and resolve conflicts between key water use sectors, considering also transboundary implications. Step 4: Inventory data and financial resources available and identify groups at risk. Step 5: Prepare/write the key tenets of the national drought management policy and preparedness plans following the three-pillar approach (early warning and prediction-pillar 1, risk and impact assessment-pillar 2, and mitigation and response-pillar 3). Step 6: Identify research needs and fill institutional gaps. Step 7: Integrate science and policy aspects of drought management. Step 8: Publicize the national drought management policy and preparedness plans, and build public awareness and consensus. Step 9: Develop education programmes for all age and stakeholder groups. Step 10: Evaluate and revise national drought management policy and supporting preparedness plans. |
| **Dealing with drought: A handbook for water suppliers in British Columbia (British Columbia, 2004 & 2022)** |



The first edition of the British Columbia Handbook was published in 2004, and the second one in 2022, which is mostly similar to the previous version. The handbook provides useful guidelines by providing specific templates for local drought management teams, drought level and response, water supply and demand assessment, drought planning and water conservation, and emergency drought planning. The main target groups are water suppliers in British Columbia, Canada. Furthermore, a drought management plan template is proposed, which is composed of eight components. Component 1: Build a local drought management team. Component 2: Document water system profile. Component 3: Evaluate the impacts of drought on the region's economy. Component 4: Monitor water supplies and climate. Component 5: Define drought stages. Component 6: Establish drought responses. Component 7: Develop communications. Component 8: Evaluate drought management plan.

**Drought risk reduction framework (UNISDR, 2007 and 2009)**

The preliminary version of the UNISDR Framework was published in 2007, and the final framework was made available in 2009. The proposed framework is composed of five core elements, each aligned to one of the five priorities of the Hyogo Framework for Action 2005-2015. Component 1: Policy and governance as an essential element for drought risk management and political commitment. Component 2: Drought risk identification, impact assessment, and early warning, which includes hazard monitoring and analysis, vulnerability and capability analysis, assessments of possible impacts, and the development of early warning and communication systems. Component 3: Drought awareness and knowledge management to create the basis for a culture of drought risk reduction and resilient communities. Component 4: Reducing underlying factors of drought risk such as changing social, economic and environmental conditions, land use, weather, water, climate variability and climate change. Component 5: Effective drought mitigation and preparedness measures to move from policies to practices in order to reduce the potential negative effects of drought.

**MEDROPLAN Guidelines (Iglesias et al., 2007)**

The MEDROPLAN Guidelines were developed under the European Commission funded MEDROPLAN project, with focus on the Mediterranean countries. The guidelines contain five main components. First, the Planning Framework to set up a multidisciplinary stakeholder team to define purpose and process. Second, Organizational Component to evaluate the legal, social and political process. Third, Methodological Component to identify risk and potential vulnerabilities. Fourth, Operational Component to identify and select both long- and short-term priority activities and actions based on the agreed criteria. Fifth, Public Review Component to conduct public review, revision and dissemination of the drought plan.

**Guidelines to develop a drought management plan for the European Union (EU) member states (European Commission, 2007)**

These EU Guidelines cover the following four key components of drought planning and management within the context of EU member states. Component 1: Drought management planning within the EU policies and river basin management plans, with focus on Water Framework Directive (WFD), and integration with the river basin development plans. Component 2: Core elements and contents of drought management plans (DMPs) including drought indicators and thresholds, measures for different phases of drought, organizational framework, and dedicated sections on dealing with prolonged droughts, and transboundary aspects. Component 3: Related issues: agriculture, groundwater and climate change. Component 4. Strategic environmental impact assessment of DMPs. Additionally, the guidelines provide examples from several EU member states to substantiate some of the key components of the proposed DMPs.

**The Near East drought planning manual (FAO and NDMC, 2008)**

The Near East Manual is tailored to the context of the Near East countries, and is mainly underpinned by the Ten-Step Process, MEDROPLAN and UNISDR guidelines. The six steps are proposed to develop and implement a national drought plan. Step 1: Creating





political momentum and authority. Step 2: Strategic planning and coordination. Step 3: Fostering involvement and developing common understanding. Step 4: Investigating drought monitoring, risk, and management options. Step 5: Writing a drought plan. Step 6: Implementing a drought plan.

**Guidelines for preparation of the drought management plans for Central and Eastern European countries (GWPCEE, 2015)**

The GWPCEE Guidelines are organized in seven main steps. These steps are largely based on the Ten-Step Process, and also consider EU and MEDROPLAN guidelines. Alignment with the relevant EU policies, especially WFD alongside the national and river basin contexts of the Central and Eastern European countries. Step 1: Develop a drought policy and establish a drought committee. Step 2: Define the objectives of a drought risk-based management policy. Step 3: Make inventory of data for the development of the drought management plan. Step 4: Produce/update the drought management plan. Step 5: Publicize drought management plan for public involvement. Step 6: Develop research and science programme. Step 7: Develop an educational programme.

**The Drought resilient and prepared Africa (DRAPA) strategic framework (UNCCD, 2018)**

The UNCCD DRAPA Framework focuses on the African context, and is composed of six elements. The proposed elements are closely related to those outlined in the UNISDR Framework. The six elements include: 1) Drought policy and governance for drought risk management; 2) Drought monitoring and early warning; 3) Drought vulnerability and impact assessment; 4) Drought mitigation, preparedness and response; 5) Knowledge management and drought awareness; and 6) Reducing underlying factors of drought risk.

**Drought hazard and risk assessment guide (World Bank, 2019)**

The World Bank Assessment Guide is organized around four main phases. Scoping phase in which issues that arise when droughts occur are broadly identified within a wider context. Inception phase in which a first estimate of the drought hazard and risk in the area of interest is made by collecting the available (relevant) data from literature as well as from a variety of other sources, in many cases online sources. Assessment phase in which a detailed analysis of ongoing, current, and/or future drought hazard and risk is carried out. Implementation phase in which actions that are most appropriate to solve the problem at hand are identified. Additionally, the guide recommends datasets, methods, models and tools that could be used in each phase. An online catalogue is also developed to support the application of these guidelines.

**Drought resilience, adaptation and management policy framework supporting technical guidelines (UNCCD, 2019)**

These UNCCD Technical Guidelines focus on the three pillars of the disaster risk reduction. The main focus under first pillar is on selecting indicators and triggers, drought forecasting system, communication and response to drought warnings, and linkages between drought risk assessment and monitoring and early warning. The pillar 2 provides guidelines to complete vulnerability and risk assessments for locations, people and economies vulnerable to drought. The pillar 3 focuses on limiting impacts of drought and better response to drought. It also delivers information on structural (physical) and non-structural measures that can be implemented to reduce the impacts of drought for nations, economic sectors and communities.

**A nine-step approach for developing and implementing an agricultural drought risk management plan (Marj and Abadi, 2020)**

The Nine-Steps for Agriculture mainly build on the existing guidelines (e.g., Ten-Step Process, EU and MEDROPLAN Guidelines, and the Near East Manual). This work proposed a tailored guide for a pilot river basin in Iran, which could be applied in other regions as well. The nine-steps also termed as phases include: Phase I: formation of the "executive team of delegations"; Phase II: encouraging stakeholders engagement; Phase III: establishing a coherent communications-network between stakeholders and teams of the plan to collaborate and exchange information; Phase IV: establishment and activation of the "recognition and assessment team"; Phase V: establishment and activation of the "supervision, monitoring, and early warning team"; Phase VI: compilation of "mitigation" and



"contingency" plans; Phase VII: activation and monitoring the "contingency" plan; Phase VIII: activation and monitoring the "mitigation" plan; Phase IX: reassessment, control, modification and updating the entire plan and sub-plans.

**A nine-step approach for participatory drought preparedness plan for hydrosystems and cities (Filho et al., 2023)**

The Nine-Steps for Hydrosystems and Cities are designed to formulate drought preparedness plans (DPPs) for hydrosystems and cities, and are underpinned by the available global knowledge and experience (especially the Ten-Step Process and the three pillars of disaster risk reduction). The approach is specifically tailored to the Brazilian context of water and drought management besides the possibility of application to other areas. This approach guides the formulation of DPPs, with first four steps resulting in a DPP without application of modelling tools (Socio-Technical DPP built mainly on the tacit knowledge), and the full nine steps including modelling approaches to facilitate in-depth scientific analysis of issues, scenarios and actions (Socio-Technical DPP with modelling intensive simulation). Step 1: Characterization of the study area. Step 2: Task force creation and initial contact with key actors attending the workshop. Step 3: Workshop 1. Step 4: Elaboration of a socio–technical drought plan. Step 5: Conducting technical visits for data collection. Step 6: Hydrological/hydraulic modeling. Step 7: Model implementation. Step 8: Conduct workshop 2 with key actors to present the results (e.g., modelling outcomes). Step 9: Final DPP–Socio–Technical with modeling-intensive simulation.

A qualitative scoring matrix was developed and used in the evaluation (Table 2). The four priority areas of the SENDAI framework along with their corresponding elements were scored at the scale of 0-100 (Very Low: 0-10; Low: 11-30; Medium-
Low: 31-50; Medium-High: 51-70; High: 71-90; and Very High: 91-100). The evaluation grid used in this study is similar to the one used for monitoring the progress on the Sustainable Development Goal 6 (SDG6) indicator 6.5 and target 6.5.1 on integrated water resources management (IWRM) (UNEP, 2021). While the scoring ranges and classes used in this study are similar to those applied for IWRM evaluation, a novel scoring grid was formulated corresponding to the objectives of this study (Table 2). Additionally, the work carried out by Fu et al., (2013), Jedd et al., (2021) and Jedd and Smith (2023) to
evaluate drought and related policies and plans was instructive in formulating the evaluation methods for this research. For the evaluation, first, each core element under a certain priority area was scored and categorized. Then, an average score was calculated for the priority area. All elements were assigned equal weights in estimating the overall score. Furthermore, a strength, weakness, opportunity and threat (SWOT) analysis was carried out. The elements scored in high to very high categories for most of the guidelines were considered as strengths. Weaknesses were identified based on elements scored under
very low to low categories in most cases. Opportunities represent areas with medium or good coverage by few guidelines, and demonstrate potential to translate into strengths with minimal efforts. Whereas, threats correspond to insufficient coverage on emerging science-policy-practice discourses in the field of disaster risk reduction, in particular, drought risk management. Ignoring or paying limited attention to these important discourses may significantly undermine the quality and effectiveness of the drought policy and planning guidelines for the future.

The evaluation results need to be interpreted with caution, owing to the inherent uncertainties associated with the evaluation process. Considering this, the overall ratings in terms of categories are used in the interpretation and discussion of the result rather than focusing on actual scores. However, the evaluation remarks alongside of scores are provided in the supplementary material for reference (Supplementary material 2). It is pertinent to note that, in some cases, the assigned scores were very





close to the border line of the two categories. These cases show comparatively higher degree of uncertainty in their
classification compared to the situations when the scores were in the middle of a category. Alongside of acknowledging these
uncertainties, it is assumed that overall pattern of scoring is likely to stay the same in most cases even if few elements are rated
bit differently within an expected uncertainty range of one category. Therefore, main patterns of the results and emerging
insights are considered reliable and instructive for further discussion, application and research by the science-policy-practice
community concerned with the drought management.

**Table 2. Description and colour scheme of the evaluation matrix developed and used in this study.**

| Classification | Score range | Scoring guide |
|---|---|---|
| Very Low (VL) | 0-10 | The element is not covered or just briefly mentioned. |
| Low (L) | 11-30 | The element is mentioned in some details, but sufficient information is lacking on concept, methods, data and tools. The references to supporting materials and examples are very limited. |
| Medium Low (ML) | 31-50 | The element is a core component of the approach. Although some information is provided on concept, methods, data and tools, important details are missing. Few references on supporting materials are included. |
| Medium High (MH) | 51-70 | The element is a core component of the approach, and receives a good coverage on concept, methods, data and tools. Most of the important details are reasonably well covered. Few references on supporting materials are included. The information is well supported by at least one or few case study examples. |
| High (H) | 71-90 | The element is a core component of the approach, and receives a very good coverage on concept, methods, data and tools. Most of the important details are well covered. Most important references on supporting materials are included and discussed in detail. The element is sufficiently underpinned by state-of-the-art on the topic and builds on the case study examples. |
| Very High (VH) | 91-100 | The element is a core component of the approach, and receives an excellent coverage on concept, methods, data and tools. The important details are covered in a comprehensive and very good manner. The element is strongly underpinned by state-of-the-art on the topic and builds on the case study examples and global best practices. |

## 3 Results and discussion

### 3.1 Performance against the SENDAI framework priorities

The evaluation results for the four priority areas of the SENDAI framework and underpinning thematic elements are presented
in Table 3. Under priority 1 (understanding the disaster risk), drought risk assessment and data and information are the two
best covered themes, which received high to very high scores for most of the evaluated guidelines. Communication and
dissemination are mostly scored in medium to high categories. In contrast, four thematic areas scored poorly in most cases.
These include local knowledge and practices, capacity development, science-policy-practice dialogue, and research and
development. These areas tend to receive lower coverage over time, as most of the guidelines developed after 2009 obtained
lower scores compared to the earlier documents. For example, science-policy-practice was well covered by the Ten-Step



Process, MEDROPLAN Guidelines and UNISDR Framework. Rest of the evaluated guidelines including most recent ones do not provide a good coverage on this topic.

Most of the evaluated guidelines scored high to very high in all thematic areas falling under disaster risk governance (priority 2). For instance, The UNISDR Framework provides a very good to excellent coverage on this priority area. Few other guidelines scoring high include: UNCCD DRAPA Framework, GWPCEE Guidelines, the Ten-Step Process, MEDROPLAN Guidelines and EU Guidelines. However, the four most recent guidelines developed during 2019-2023 (World Bank Assessment Guide; UNCCD Technical Guidelines, Nine-Steps for Agriculture, and Nine-Steps for Hydrosystems and Cities) obtained comparatively low scores. These guidelines scored very low to low for political will; low to medium low for periodic assessment and reporting; and medium-low to medium-high for policy and governance aspects. The most recent guidelines place very high emphasis on covering three-pillars of drought risk reduction, and tend to give less attention to other important themes linked to the SENDAI framework. In contrast, drought risk reduction strategies and plans received good to excellent coverage by the evaluated guidelines. Similarly, stakeholder participation including community engagement, and coordination mechanisms within or across multiple sectors are very well covered in most cases.

The scores for priority 3 (investing in disaster reduction for resilience) were very low to low in most cases. Only one of the twelve drought guidelines, UNISDR Framework, scored in medium-high to high range for the key elements under priority 3. Rest of the eleven guidelines mostly achieved (very) low to medium scores. For example, resource allocation (especially finance) and risk transfer (including insurance) are either not a core element or lack sufficient coverage in most cases. Similarly, mainstreaming drought risk reduction into land use policies and rural development plans lacked sufficient attention. Business resilience and protection of livelihoods and productive assets, and health and safety are classified under very low to low categories because of insufficient coverage. However, sustainable use and management of ecosystem received variable coverage, as few guidelines (UNISDR and UNCCD DRAPA Frameworks; EU and GWPCEE Guidelines) provide a good to very good coverage on this theme. Last but not least, most thematic areas under priority 4 are rated under low to medium categories. An exception is the topic of disaster preparedness and contingency policies, plans and programs, which received medium to high coverage in most cases. Whereas, least attention is paid to elements related to post-disaster recovery, rehabilitation and reconstruction; resilience of critical infrastructure; and multi-hazard forecasting and early warning systems.

**Table 3. The evaluation results on the alignment of the examined guidelines with the four priority areas of the SENDAI Framework**

| SENDAI framework priority area, and main elements considered in the evaluation | The Ten-Step Process | British Columbia Handbook | UNISDR Framework | MEDROPLAN Guidelines | EU Guidelines | The Near East Manual | GWPCEE Guidelines | UNCCD DRAPA Framework | World Bank Assessment Guide | UNCCD Technical Guidelines | Nine-Steps for Agriculture | Nine-Steps for Hydrosystems and Cities |
|---|---|---|---|---|---|---|---|---|---|---|---|---|
| **"Priority 1 Understanding disaster risk.** Disaster risk management needs to be based on an understanding of disaster risk in all its dimensions of vulnerability, capacity, exposure of persons and assets, hazard characteristics and the environment." (UNISDR, 2015) | | | | | | | | | | | | |
| **Overall evaluation priority 1** | MH | ML | H | MH | MH | ML | MH | MH | ML | ML | ML | L |
| Data and information | H | H | VH | H | H | H | H | VH | VH | VH | MH | MH |
| Risk assessment | H | MH | VH | VH | H | H | H | MH | VH | VH | MH | MH |




| SENDAI framework priority area, and main elements considered in the evaluation | The Ten-Step Process | British Columbia Handbook | UNISDR Framework | MEDRO PLAN Guidelines | EU Guidelines | The Near East Manual | GWPCEE Guidelines | UNCCD DRAPA Framework | World Bank Assessment Guide | UNCCD Technical Guidelines | Nine-Steps for Agriculture | Nine-Steps for Hydrosystems and Cities |
|---|---|---|---|---|---|---|---|---|---|---|---|---|
| Local knowledge and practices | VL | L | H | VL | VL | VL | VL | L | VL | VL | VL | VL |
| Capacity Development | H | L | VH | ML | ML | VL | ML | MH | L | L | L | VL |
| Science-policy-practice dialogue | H | VL | H | H | VL | VL | VL | L | VL | VL | VL | VL |
| Research and development | MH | VL | MH | L | H | L | MH | L | L | L | ML | VL |
| Communication and dissemination | H | MH | VH | MH | H | L | H | MH | MH | H | MH | ML |
| **"Priority 2 Strengthening disaster risk governance to manage disaster risk.** Disaster risk governance at the national, regional and global levels is vital to the management of disaster risk reduction in all sectors and ensuring the coherence of national and local frameworks of laws, regulations and public policies that, by defining roles and responsibilities, guide, encourage and incentivize the public and private sectors to take-action and address disaster risk." (UNISDR, 2015) | | | | | | | | | | | | |
| **Overall evaluation priority 2** | **H** | **MH** | **VH** | **H** | **H** | **H** | **H** | **H** | **MH** | **ML** | **MH** | **ML** |
| Policy and governance | H | ML | VH | H | H | MH | H | VH | MH | ML | ML | ML |
| Strategies and plans | H | H | VH | H | 85 | MH | H | H | H | H | H | H |
| Community representation | MH | H | VH | MH | MH | MH | MH | H | H | H | H | ML |
| Coordination mechanisms | H | H | VH | H | H | H | H | H | MH | MH | H | H |
| Political will and support | H | H | VH | H | H | H | H | H | L | VL | L | VL |
| Periodic assessment and reporting | H | ML | H | H | H | MH | H | MH | L | VL | ML | ML |
| **"Priority 3 Investing in disaster risk reduction for resilience.** Public and private investment in disaster risk prevention and reduction through structural and non-structural measures are essential to enhance the economic, social, health and cultural resilience of persons, communities, countries and their assets, as well as the environment. These can be drivers of innovation, growth and job creation. Such measures are cost effective and instrumental to save lives, prevent and reduce losses and ensure effective recovery and rehabilitation." | | | | | | | | | | | | |
| **Overall evaluation priority 3** | **ML** | **L** | **H** | **L** | **ML** | **L** | **L** | **ML** | **ML** | **ML** | **L** | **L** |
| Resource allocation including finance | MH | VL | H | MH | VL | VL | L | MH | MH | ML | VL | VL |
| Risk transfer and insurance | L | VL | H | MH | VL | L | 10 | MH | ML | H | VL | VL |
| Mainstreaming Disaster risk reduction assessments into land use policy | L | ML | H | L | L | L | L | L | ML | MH | ML | VL |
| Mainstreaming disaster risk reduction into rural development plans | L | L | MH | VL | H | L | L | L | L | L | VL | VL |
| Business resilience and protection of livelihoods and productive assets | ML | L | H | L | L | L | VL | ML | ML | L | ML | VL |
| Sustainable use and management of ecosystems | ML | ML | H | ML | H | ML | H | H | MH | MH | L | MH |
| Health and safety | L | L | MH | L | L | L | L | MH | L | ML | VL | L |
| **"Priority 4 Enhancing disaster preparedness for effective response, and to «Build Back Better» in recovery, rehabilitation and reconstruction.** Experience indicates that disaster preparedness needs to be strengthened for more effective response and ensure capacities are in place for effective recovery. Disasters have also demonstrated that the recovery, rehabilitation and reconstruction phase, which needs to be prepared ahead of the disaster, is an opportunity to «Build Back Better» through integrating disaster risk reduction measures. Women and persons with disabilities should publicly lead and promote gender equitable and universally accessible approaches during the response and reconstruction phases." (UNISDR, 2015) | | | | | | | | | | | | |
| **Overall evaluation priority 4** | **ML** | **ML** | **MH** | **ML** | **MH** | **ML** | **ML** | **MH** | **ML** | **ML** | **ML** | **ML** |
| Disaster preparedness and contingency policies, plans and programmes | H | H | H | H | H | H | H | H | MH | L | H | MH |
| People-centred multi-hazard, multisectoral forecasting and early warning systems | ML | ML | ML | ML | ML | ML | MH | ML | MH | ML | L | ML |





| SENDAI framework priority area, and main elements considered in the evaluation | The Ten-Step Process | British Columbia Handbook | UNISDR Framework | MEDRO PLAN Guidelines | EU Guidelines | The Near East Manual | GWPCEE Guidelines | UNCCD DRAPA Framework | World Bank Assessment Guide | UNCCD Technical Guidelines | Nine-Steps for Agriculture | Nine-Steps for Hydrosystems and Cities |
|---|---|---|---|---|---|---|---|---|---|---|---|---|
| Disaster response including in emergencies | MH | MH | MH | MH | ML | L | ML | MH | ML | ML | L | L |
| Post-disaster recovery, rehabilitation and reconstruction | ML | L | H | L | ML | L | L | ML | L | L | L | L |
| Resilience of new and existing critical infrastructure | L | L | MH | ML | MH | L | VL | L | ML | ML | L | L |

## 3.2 Overall assessment and SWOT analysis

Figure 1 shows the average ratings of the examined guidelines against each of the four priority areas of the SENDAI
Framework. In general, none of the examined guidelines align very well with all the four priority areas of the SENDAI
framework. Nevertheless, UNISDR Framework performed better compared to other guidelines examined in this study, even
though it needs considerable improvement on priority 3 and 4. Contrary to the expectation, the most widely adopted Ten-Step
Process could not score very high on any of the four priority areas, but scored Medium-Low in two of the four priorities (3&4),
Medium-High for priority 2, and High for priority 1. Couple of the examined guidelines (UNCCD Technical Guidelines and
World Bank Assessment Guide) are focused on few thematic areas such as addressing the three pillars of disaster risk reduction,
and scored high to very high under these elements, but achieved low to medium overall scores for all the four priorities. On
the other hand, the two most recent guidelines (Nine-Steps for Agriculture, and Nine-Steps for Hydrosystems and Cities)
aiming to provide a comprehensive drought planning process also achieved lower scores in general. Similarly, the regional
guideless (The Near East Manual, UNCCD-DRAPA Framework, and MEDROPLAN and EU Guidelines) achieve low to
medium scores in most cases. Furthermore, building on these evaluation results, the SWOT analysis was conducted, which is
summarized in Fig. 2, and discussed below.





Figure 1: Average scores obtained by the examined drought guidelines for each of the four priority areas of the SENDAI framework.





**Strengths**

1. Data and information
2. Risk assessment
3. Policy and governance
4. Strategies and plans
5. Community representation
6. Coordination mechanisms

**Opportunities**

1. Capacity development
2. Communication and dissemination
3. Research and development
4. Political will and support
5. Periodic assessment and reporting
6. Resource allocation including finance
7. Risk transfer and insurance
8. Sustainable use and management of ecosystems
9. Disaster preparedness, contingency policies , plans and programmes

**Weaknesses**

1. Local knowledge and practices
2. Science-policy-practice dialogue
3. Mainstreaming disaster risk reduction into land use policies and rural development plans
4. Business resilience and protection of livelihoods
5. Health and safety
6. Multi-hazard approach
7. Disaster response including in emergencies
8. Post-disaster recovery
9. Resilience of critical infrastructure

**Threats**

1. Lack of alignment with global disaster risk reduction agenda
2. Increasing trend towards reductionism (e.g., too much emphasis on three pillars of drought risk management)
3. Slow transition towards risk management
4. Lack of guidance on crisis management
5. Lack of periodic updates on the guidelines
6. Lack of correspondence with emerging science-policy-practice discourses

**Figure 2: The summary of SWOT analysis conducted on the examined drought guidelines**

**3.2.1    Strengths**

There are several areas which are covered (very) well by most of the guidelines (strengths) including data and information, risk assessment, policies and plans, coordination and stakeholder participation (Fig. 2). These areas should be kept during new developments, updates or applications, as these subjects will require little to moderated efforts to adjust to the scope and context of the new guidelines. The available guidelines provide a detailed account on the state-of-the-art on these topics, which

can be very instructive for the future work. For example, drought risk assessment is very well covered by the Ten-Step Process, MEDROPLAN Guidelines, UNISDR Framework, UNCCD Technical Guidelines, and World Bank Assessment Guide. The concepts, methods and data for assessing drought hazard, exposure, impact and coping capacity are very well explained in most of these documents. Moreover, combining various factors in assessing drought vulnerability and risk are clearly outlined. These guidelines also provide a good to very good coverage on the aspects related to data and information, policies and plans,

coordination and stakeholder participation, therefore, can serve as an excellent reference for future work on updating the guidelines or applying them in practice.



### 3.2.2 Weaknesses

Nine areas were identified as weaknesses (Fig. 2), which require urgent attention. Making progress on these areas will require an inquiry beyond the available drought guidelines, which provide limited information on these aspects. For example, the examined drought guidelines lacked good coverage on people-centred multi-hazard, multisectoral forecasting and early warning systems. While multiple sectors impacted by drought are mentioned, tailoring of early warning systems to cater the needs of various sectors is not yet well developed and remain poorly covered. Additionally, the examined guidelines lacked sufficient focus on establishing linkages of drought and other natural or manmade hazards such wildfires, heatwaves, desertification, water scarcity and floods. However, the available scientific research and some practice documents can contribute in converting outlined weaknesses to strengths. For example, available literature can be helpful on understanding the linkages between drought and other hazards such as drought and desertification (Stringer et al., 2009; UNCCD, 2022; Oswald and Harris, 2023), floods and droughts (Ward et al., 2020; Browder et al., 2021) and compound events in general (Zscheischler et al., 2018). Similarly, available studies can provide useful guidance on strengthening information on multi-hazard early warning systems (e.g., Aguirre-Ayerbe et al., 2020; Hemachandra et al., 2021; UNDRR and WMO, 2023).

### 3.2.3 Opportunities

Seven opportunities were identified (Fig. 2). These areas can be enhanced, capitalizing on the information already available in the examined guidelines. For example, UNISDR Framework provides a good description on investments for prevention, mitigation and preparedness measures underpinning them by examples and references to various investment sources. Similarly, the Ten-Step Process and UNCCD DRAPA Framework recommends innovative financial mechanisms alongside funding from various sources such as public and private investments. Whereas, MEDROPLAN Guidelines, UNISDR Framework, and UNCCD Technical Guidelines contain some useful insights on risk transfer, and insurance and safety nets alongside of few good examples. Additional insights from multiple sources could provide useful material to strengthen future drought guidelines on these aspects (see for example, Tadesse et al., 2015; Kron et al., 2016; World Bank, 2022; World Bank and European Commission, 2024).

### 3.2.4 Threats

The major threats include: lack of alignment with global disaster risk reduction agenda; increasing trend towards reductionism; slow transition towards risk management; lack of guidance on crisis management; and lack of periodic updates on the guidelines (Fig. 2). For example, to date, there is no guideline specifically designed to align with the contemporary science-policy-practices discourses and global disaster risk reduction agenda-the SENDAI Framework. Only UNISDR Framework was drafted in response to the Hyogo framework for action 2005-2015; hence, aligns very well to its priority areas, but requires significant update on two of the four priority areas of the SENDAI framework (priority 3 & 4). Furthermore, too much



emphasis on risk management may be counterproductive, as the focus on crisis management receive no or little attention. This is demonstrated by the weak coverage on disaster response including in emergencies and post-disaster recovery, rehabilitation and reconstruction. None of the evaluated guidelines provide a comprehensive coverage on the key elements pertaining to the crisis management. Nevertheless, one of the twelve guidelines, British Columbia Handbook (2004) contains useful information on emergency response planning. Last but not least, available guidelines lacked correspondence with the contemporary research and development discourses, and can benefit from the available literature on these areas. Examples include, but not limited to: understanding drought in the Anthropocene (Van Loon et al., 2016 & 2022, Cook et al., 2022; Hall et al., 2022), transition to sustainability and achieving SDGs where drought management is an important contributor (Zhang et al., 2019; UNDRR, 2022; Tabari and Willems, 2023); assessing climate change impacts and adaptation options (Stringer et al., 2009; Cook et al., 2018; C2ES, 2018; Dai et al., 2018; Mukherjee et al., 2018; Iglasias et al., 2021), addressing maladaptation (Christian-Smith et al., 2015; Ward et al., 2020; Filho et al., 2022; Rechien et al., 2023; Tubi and Israeli, 2024); and managing the risk from flash (Otkin et al., 2018; Christian et al., 2021; Yuan et al., 2023) and mega droughts (Gober et al., 2016; Garreaud et al., 2020; Cook et al., 2022).

### 3.3 Making transition towards next generation of drought policy and planning guidelines

Since available drought policy and planning guidelines do not align very well with the contemporary disaster risk reduction agenda, there is an urgent need to revise and improve them or develop new guidelines. This is essential to accelerate progress on the transition towards risk management, building resilience and sustainability. At a global level, efforts could be dedicated to revisit the available guidelines. For example, UNISDR Framework could be improved to better align with SENDAI framework priorities. Moreover, the Ten-Step Process could be updated, as it is very valuable and most widely recommended drought guide, but has not been significantly updated since the work of Wilhite et al. (2000). Similarly, regional or local guidelines need considerable improvements in several areas. At one hand, it is recognized that some guidelines may have been resulting from dedicated projects, and it may be difficult to revisit them after the project closures. On the other hand, like policies and plans need periodic evaluation and revision, so are the guidelines underpinning them. Thus, the drought guidelines are not meant to be static. Therefore, it is highly recommended to make concerted efforts at global, regional and local levels to dynamically update the guidelines so that these are corresponding well with the contemporary thinking and changing needs. There are several institutions and groups (e.g., UN agencies; academia, research groups, donors, and public and private sector organizations) who can (naturally) paly a leading role in taking up this urgent call, as these institutions have a mandate, and made significant contributions on guiding drought policy, planning and practical implementation in the past.

The information presented in this research can provide useful insights for both developers and the users of the drought guidelines to move towards next generation of drought policy and planning guidelines. It is recognized that developing guidelines require large investments and collaborative efforts from multiple stakeholders. Therefore, developing new or updated guidelines is beyond the scope of this research. Nevertheless, a contemporary framework is provided to facilitate this





process (Fig. 3). The proposed framework is underpinned by the valuable information available through the existing guidelines, and new insights generated from this study. The framework contains seven main steps and few cross-cutting elements linked to each step. Additionally, the process steps, potential thematic elements, and linkages with the SENDAI framework priorities are briefly mentioned in Table 4. In general, the proposed framework is flexible and could be adopted to the users' needs, for example, by adding another step, a cross-cutting element, or by establishing linkages with relevant global, regional and local policies.

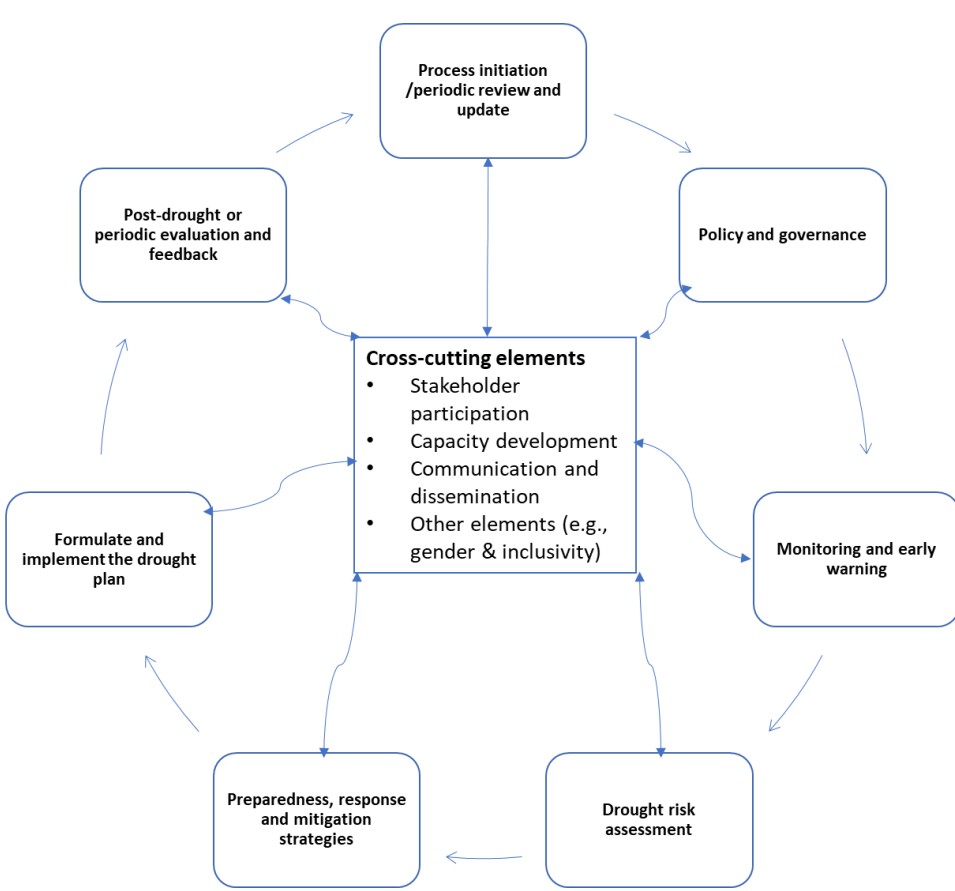

**Figure 3. Schematic representation of the key steps and elements for supporting the development of next generation of drought policy and planning guidelines**

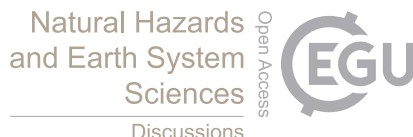

**Table 4. A template enumerating the key steps and elements for supporting the development of next generation of drought policy and planning guidelines.**

| Process step | Suggested elements | Link to SENDAI framework priorities |
|---|---|---|
| Process initiation or periodic review and update | • Triggers (e.g., changing conditions and needs; drought event)<br>• Periodic review and update as part of regular planning cycle<br>• Commitment from relevant authorities<br>• Identification of the leading authorities, organizations and teams<br>• Add any other points | Priority 2 |
| Policy and governance | • Forming drought policy and governance authority/commission; if necessary also form the lead teams/committees/groups<br>• Analyzing existing policy and governance arrangements related to drought<br>• Analyzing existing policy and governance arrangements related to other natural or man-made hazards, or natural resources management, NRM, (water management, land use and forestry, environment, climate change etc)<br>• Assessing organizational structure (e.g., public and private sector organizations) for drought, other hazards or NRM management<br>• Assessing policy coherence, multi-hazard and cross sectoral coordination<br>• Formulate new or revised policy and governance arrangements<br>• Add any other points | Priority 2 |
| Drought monitoring and early warning | • Monitoring and reporting on different drought types (e.g., meteorological, agricultural, hydrological, socio-economic drought) across relevant spatial (e.g., global to local, river basin to small catchment) and temporal scales (short-term/flash droughts; monthly, seasonal, annual or multi-year) including consideration of flash and mega droughts<br>• Monitoring and reporting on the linkages of drought and other natural or man-made hazards (e.g., heatwaves, wildfires, water scarcity, desertification, and floods)<br>• Drought forecasting and early warning as part of multi-hazard early warning system<br>• Climate change impact assessment and plausible future scenarios<br>• Integrate scientific and local knowledge (where appropriate)<br>• Formulate committees/groups spearheading the work on drought monitoring and early warning<br>• Add any other points | Priority 1 |
| Drought risk assessment | • Assessing impacts of drought (and linked hazards) by taking multi-sectoral approach as well as including vulnerable communities and ecosystems<br>• Assessing vulnerability to drought underpinned by exposure, impact and coping capacity analyses<br>• Assessing and mapping the risk of drought (and other multiple/linked hazards) using state-of-the art methods (e.g., by combining hazard and vulnerability assessments)<br>• Integrate scientific and local knowledge (where appropriate)<br>• Formulate committees/groups spearheading the work on drought risk assessment<br>• Add any other points | Priority 1 |



| | | |
|---|---|---|
| Preparedness, response and mitigation strategies | • Select suitable preparedness and mitigation measures to reduce drought risk<br>• Select suitable measures to respond during different stages of droughts (e.g., pre-alert, alert and emergency or other classification used in a specific context)<br>• Examine preferred measures and strategies for adaptive planning (e.g., flexibility and robustness to address deep uncertainty) considering scenarios for climate change and anthropogenic developments and pressures in the future<br>• Examine preferred measures and strategies to avoid maladaptation<br>• Examine preferred strategies for contribution to achieving the local and global disaster risk reduction and sustainable development agenda (e.g., SENDAI framework goals and objectives; contribution towards drought related SDGs and targets)<br>• Apply multi-criteria analysis to perform comprehensive evaluation of the proposed strategies<br>• Co-develop and co-evaluate the plausible strategies with relevant stakeholders (including women and most vulnerable groups) and decision makers, and revise where necessary to get them approved by the relevant authorities<br>• Formulate committees/groups spearheading the work on drought preparedness, response and mitigation strategies<br>• Add other points | Priority 4 |
| Formulate and implement drought plan | • Draft drought plan including policy, governance, drought risk, and preferred preparedness, response (including in emergencies) and mitigation strategies<br>• Outline implementation aspects such as resources allocation including finance, institutional roles and responsibilities, time-frame and (sequence of) measures under pre-and-post drought situation such as pre-alert, alert and emergency situations<br>• Discuss the drought plan with relevant stakeholders and decision makers, and revise where necessary to get it approved by the relevant authorities<br>• Publicize the drought policies and plans<br>• Add any other points | Priority 3 |
| Post-drought or periodic evaluation and feedback | • Regularly monitor and evaluate the implementation of drought policies and plans<br>• Conduct special post-drought evaluations after every drought event<br>• Provide feedback to improve drought policy and plans<br>• Add any other points | Priority 1, 2 & 4 |
| Cross-cutting elements | • Select cross-cutting elements including but not limited to:<br>• Stakeholder participation<br>• Capacity development<br>• Communication and dissemination<br>• Add any other cross-cutting elements (e.g., gender and inclusivity) | May cover Priority 1-4 |

**4 Conclusions and recommendations**

A number of drought policy and planning guidelines have been developed and used over the last few decades. However, there

is a lack of understanding on the alignment of these guidelines with the contemporary disaster risk reduction agenda. This



study evaluated twelve drought policy and planning guidelines for their alignment with the four priority areas of the SENDAI Framework for disaster risk reduction 2015-2030. The study shows that the available guidelines stress the need of a transition from crisis to risk management. However, despite providing useful instructions, transition towards risk management and building resilience is still a global challenge. While global disaster risk reduction agendas have attempted to keep pace with addressing emerging challenges, the drought policy and planning guidelines have not sufficiently responded to these new developments.

This study concludes that the current drought guidelines do not aligned very well with the contemporary disaster risk reduction agenda. While the available guidelines do provide very valuable instructions on several important areas (e.g., data and information; risk assessment; coordination mechanism and stakeholder participation; policy and governance; preparedness plans; and communication and dissemination), there are a number of key elements necessitating substantial improvement (e.g., local knowledge and practices; resource allocation including finance; risk transfer and insurance; mainstreaming drought risk reduction into land use and rural development policies; post-disaster recovery; rehabilitation and reconstruction, business resilience and protection of livelihoods; health and safety; resilience of critical infrastructure; and science-policy-practice dialogue). The drought policy and planning guidelines need periodic revisions to remain valid to address the contemporary challenges and needs. Therefore, it is recommended to update the drought guidelines after every ten to fifteen years in the light of new developments in the relevant agendas and scientific knowledge. Finally, this research calls for an urgent and overdue action to make concerted efforts in developing next generation of drought policy and planning guidelines. The wealth of information available through previous work and new insights presented in this study can substantially contribute in these developments supporting the accelerated transition towards improved drought risk management, and building resilience of societies and ecosystems against droughts under changing climate and increasing anthropogenic pressures.

**Data availability**

The study is based on the document analysis, and the references are provided for the examined sources.

**Author contribution**

Ilyas Masih, as the single author of this manuscript, contributed to all phases of the manuscript development including conceptualization, literature search and selection of the drought policy and planning guidelines for an in-depth analysis, development of evaluation methodology, conducting the analysis, and manuscript writing.

**Competing interests**

The Author declares no competing interests.



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
