# Peer review of "An evaluation on the alignment of drought policy and planning guidelines with the contemporary disaster risk reduction agenda"

_Natural Hazards and Earth System Sciences, 2024_

## Author Response (AR2)

**Author's response to the comments made by the reviewers on the manuscript, "Masih, I.: An evaluation on the alignment of drought policy and planning guidelines with the contemporary disaster risk reduction agenda, Nat. Hazards Earth Syst. Sci. Discuss. [preprint], https://doi.org/10.5194/nhess-2024-163, in review, 2024."**

Reviewer 1 (Winnie Khaemba):

Dear Reviewer,

Thank you very much for your time and effort in reviewing this manuscript, and providing your valuable comments. These are highly valued and much appreciated. Please see below the point-by-point reply to your review comments and how these are addressed in the revised manuscript.

**Review comment:** Overall this is a good manuscript with some really interesting insights on the guidelines related to drought response.

**Response:** Thank you very much for your kind words of appreciation. These are encouraging and add to the confidence and motivation to present and further develop this work.

**Review comment:** Risk reduction versus risk management: with Sendai coming into place their has been more reference to risk management as opposed to risk reduction and I see this manuscript is framed around risk reduction so it would be interesting to highlight the alignment with the current discussions on risk management in a more explicit way.

**Response:** Thank you very much for your valuable comment. The terminology for risk reduction and management is clarified in the revised manuscript under methodology section. Moreover, the document is strengthened in detailing risk management strategies and their implementation aspects aimed at reducing drought risks (see revisions made in section 3.2: overall assessment and SWOT analysis).

**Review comment:** Context: It would be very helpful to provide some context for the guidelines selected for analysis. This helps in providing some background on why they score the way they do.

**Response:** This is a very helpful comment for the benefit of the readers who are not familiar with drought policy and planning guidelines. More information is added in Table 1 on the context and background of the examined guidelines.

**Review comment:** Scoring scale: The scoring scale (it would be good to provide an explanation on the range which is very wide - maybe would have been better to have 1-5 or 1-10 as the range to better capture the nuance as the long range does not reflect in the results thus there is failure to convey the nuances within the range (e.g 11-30 - an area with 12 and another at 29 are still grouped together with no further breakdown). The UNISDR framework ran until 2021 (2016-2021) - so possibly mention this context.

**Response:** Thank you very much for your critical insight on the scoring scale. After careful consideration of different options, it is decided to keep the same scoring scale and description outlined in Table 2. The used approach is well established and provide an opportunity to compare the results within one category. Moreover, it is assumed that changing the scoring scale would not have significant impact on the main findings and conclusions of this research. Therefore, it is preferred not to change the evaluation scale and corresponding grading system, which would essentially require a re-assessment of the whole work. Nevertheless, the background information on the evaluated guidelines is strengthened (Table 1) and analysis and discussion of results is enhanced

(Section 3.2 on overall assessment and SWOT analysis). These improvements will facilitate the reader in better understanding the methods and results.

Moreover, as per recommendation, the context of UNISDR strategic framework 2016-2021 is mentioned in the introduction section, highlighting the contribution of disaster risk reduction to achieving the sustainable development agenda.

**Review comment:** Multi-hazards: What about multi-hazards? it is important to have reflections around this as there is quite some focus on this with climate extremes becoming more frequent and intense thus need for a more comprehensive approach in dealing with hazards and I believe this submission should include this aspect.

**Response:** Indeed, dealing with multiple hazards is very important, especially in the light of observations and future predictions indicating climate extremes becoming more frequent and intense. Reflections around multi-hazards is strengthened in the revised manuscript (section 3.2.2: Weaknesses).

**Reviewer 2 (Ana Iglesias)**

The manuscript reflects of the need to update of the available drought guidelines to include the contemporary disaster risk reduction agenda. The evaluation is rigorous and well presented, including a schematic representation of the key elements for supporting the development of next generation of drought policy and planning guidelines.

However, the priorities given to each of the elements seem to be abstract since they are not linked to realistic implementation. For example, priority 1 is given to the technical aspects (i.e., drought monitoring and early warning and drought risk assessment), while the more social and cultural aspects of the process are given less priorities (i.e., policies, preparedness and mitigation strategies).

In my view, recent examples in disaster risk management highlight the critical importance of these social and cultural elements in supporting the transition towards disaster management for resilient societies.

It will be interesting if the manuscript could provide a brief discussion on how the social and cultural aspects of drought management and the more technical aspects are mutually supportive and essential to build resilient societies.

**Response:**

Thanks for spending your valuable time and efforts in reviewing the manuscript and providing very helpful comments. The revised document provides more information and discussion on the linkages and synergies between social and technical aspects (see for example changes made under Section 3.2 on overall assessment and SWOT analysis, and sub-section 3.2.2: Weaknesses; where the importance of systemic approach is highlighted including social and technical aspects).